# Pre-screening workers to overcome bias amplification in online labour markets

**Ans Vercammen**[1]*, **Alexandru Marcoci**[2], **Mark Burgman**[1]

**1** Centre for Environmental Policy, Imperial College London, London, United Kingdom, **2** Philosophy Department, UNC Chapel Hill, Chapel Hill, NC, United States of America

* Ans.Vercammen15@imperial.ac.uk

## Abstract

Groups have access to more diverse information and typically outperform individuals on problem solving tasks. Crowdsolving utilises this principle to generate novel and/or superior solutions to intellective tasks by pooling the inputs from a distributed online crowd. However, it is unclear whether this particular instance of "wisdom of the crowd" can overcome the influence of potent cognitive biases that habitually lead individuals to commit reasoning errors. We empirically test the prevalence of cognitive bias on a popular crowdsourcing platform, examining susceptibility to bias of online panels at the individual and aggregate levels. We then investigate the use of the Cognitive Reflection Test, notable for its predictive validity for both susceptibility to cognitive biases in test settings and real-life reasoning, as a screening tool to improve collective performance. We find that systematic biases in crowdsourced answers are not as prevalent as anticipated, but when they occur, biases are amplified with increasing group size, as predicted by the Condorcet Jury Theorem. The results further suggest that pre-screening individuals with the Cognitive Reflection Test can substantially enhance collective judgement and improve crowdsolving performance.

**Data Availability Statement:** The study was registered on the Open Science Framework website (https://osf.io/7d69p/) and the data and materials are made available there.

## Introduction

Empirical evidence supports the idea that the aggregate judgement of diverse and independent contributors typically outperforms the assessment of (even a knowledgeable) individual, commonly referred to as the wisdom-of-the-crowd effect [1, 2]. With increased online connectivity, those looking to solve a problem, or obtain a collective judgement can now tap into large, increasingly diverse and readily accessible panels of online workers, securing answers effectively and expediently. In recent years, this technique has been applied to crowdsource geopolitical predictions [3], medical decision-making [4], stock price forecasting [5], policy making [6], meteorological forecasts [7, 8], investigative journalism [9] and citizen science [10].

Crowdsourcing platforms (e.g. Amazon's Mechanical Turk; AMT) are online labour markets, offering a range of microtasks or larger jobs. Academic researchers are increasingly turning to crowdsourcing platforms and other readily available online panels (e.g. Qualtrics, Prolific) to access survey respondents or to conduct online experiments [11, 12]. Online workers typically self-select the tasks they wish to complete [13]. Task selection based on the

**Funding:** The research reported herein is partly based upon work supported by the Office of the Director of National Intelligence (ODNI), Intelligence Advanced Research Projects Activity (IARPA), under contract (2017-16122000002). Views and conclusions herein are those of the authors and should not be interpreted as necessarily representing the official policies, either expressed or implied, of ODNI, IARPA or the United States Government. The United States Government is authorized to reproduce and distribute reprints for governmental purposes notwithstanding any copyright annotation therein.

**Competing interests:** The authors have declared that no competing interests exist.

worker's preference should enhance intrinsic motivation and expertise in the specific task, however, workers seem to lack accurate awareness of their level of competency [14]. Furthermore, task selection may not be systematic. For instance, fresh tasks on AMT are almost 10 times more attractive for workers than older tasks [15]. The effort required to search for suitable tasks (in terms of a workers' competencies or interests), or in some cases a lack of alternatives [16] may lead to poor worker-task matchups and loss of quality. Overall, the abundance of low-quality work harms the reliability, scalability, and robustness of online labour markets [17].

Efforts to mitigate quality loss in online labour markets have generally focused on strategic implementation of incentives, performance tracking (e.g. AMT maintains a worker reputation measure based on how many pieces of their work were accepted by past requesters), or by incorporating 'attention check questions' [18–20]. These strategies can aid requesters to selecting highly motivated participants and eliminate spammers and malicious responders in a post-hoc manner. However, they do not alleviate unintentional performance deficits due to cognitive biases which may lead to poor outcomes despite honest effort. What is more, individual decisions affected by cognitive bias may compound to decouple the relationship between wisdom (i.e., the quality of ideas) and crowds (i.e., their popularity) [21]. As micro-tasking, an ostensibly straightforward form of crowdsourcing, is affected by cognitive bias [22], one might also expect 'crowdsolving', which relies on workers' reasoning ability to solve complex problems, to be impacted by lapses in rationality.

To understand the implications of bias in crowdsourced solutions, particularly those that use majority voting, it is important to note that the outcome of response aggregation is expected to follow Condorcet's Jury Theorem [23] on both multiple choice and open-ended tasks [24]. Assuming voters have an independent probability p of making the correct decision, the theorem states that in the case where p is larger than .5, increasing the group size also increases the probability that the majority vote will be correct. In the case where there are more than two possible responses, we know from a generalization of the Condorcet Jury theorem that:

> The epistemically correct choice is the most probable among k options to be the plurality winner, just so long as each voter's probability of voting for the correct outcome exceeds each of that voter's probabilities of voting for any of the wrong outcomes. This implies that, if error is distributed perfectly equally, a better than 1/k chance of being correct is sufficient for the epistemically correct option to be most likely to be the plurality winner among k options ([25], p. 286).

It follows that one might expect nominal groups to provide the correct answer if the population from which the group is drawn favours the correct answer. Conversely, and given the fact that individuals regularly violate normative standards in reasoning [26], systematic reasoning errors may be so prevalent that an incorrect answer would command a relative majority, leading to the selection of an erroneous outcome in a nominal group setting. To date, with some notable exceptions [4, 24], empirical verification is lacking on the extent to which common aggregation techniques in online samples could enhance reasoning performance or, alternatively, amplify common reasoning errors. Furthermore, one issue of practical relevance remains poorly understood, namely, what effect nominal group size has on the accuracy of crowdsolved problems.

Given the prevailing evidence of the widespread nature of cognitive biases in individuals, and the implications of the Condorcet Jury Theorem on the aggregation of judgements, it follows that crowdsolving would suffer the detrimental effects of common failures in individual

reasoning. However, these assumptions remain untested in online labour markets. This study therefore addresses two unresolved issues. First, it provides an empirical examination of the extent to which typical online labourers recruited via a popular online panel service commit reasoning errors, and how this affects the aggregate performance of nominal groups in crowd-solving challenges. Second, as the literature suggests that there are substantial individual differences in the extent to which individuals enjoy engaging in effortful thinking [27] and reasoning errors [28, 29], we examine whether the short Cognitive Reflection Test (CRT) could be used as a screening tool to identify high-performing workers and enhance crowd-sourced outputs. While the CRT has been convincingly shown to predict performance on the heuristics and biases problems used in this study [28, 29], to our knowledge, it remains to be demonstrated whether the CRT is associated with susceptibility to cognitive bias in typical online panels. There are reasons to be cautious about the transferability of results obtained in student and community samples to these novel, purposely designed online labour markets [30, 31]. Therefore, our second objective was to determine to what extent (if at all) prior reasoning performance-based pre-selection can mitigate against bias inflation in a crowdsourced sample.

## Materials and methods

All procedures were approved by the Imperial College Research Ethics Committee (approval number 17IC4226). The study was registered on the Open Science Framework website (https://osf.io/7d69p/), where a copy of all materials and data can be found. The original aim of the study was solely to examine the effect of group size on reasoning performance, following on from previous research [32, 33] showing that inductive reasoning, as measured with the Raven's Standard Progressive Matrices, rapidly improves with the size of the group. Here we report on additional exploratory analyses that focus on the utility of the concept of Cognitive Reflection, and its operationalisation in a short test (the CRT), to screen workers and enhance the quality of reasoning in crowdsourced outputs.

### Participants

We recruited participants through Qualtrics Research Panels, a service that provides access to representative samples of the population by accessing a range of different panel agencies. Data were obtained from a total of 105 participants, but pilot data from the first 10 participants were discarded due to issues with data quality. For the remaining 95 participants, basic quality control mechanisms were implemented. We included a simple attention check to ensure that participants were mindful, and as per standard procedure all participants received a cash reward (US $8.50) credited to their member account, redeemable for a gift card. We imposed no location restrictions, but participants had to be at least 18 years old and have at least intermediate English language proficiency, ascertained through screening questions as part of the consent procedure. All participants gave prior informed consent.

Our sample had an average age of 48.61 years (SD = 15.41). Forty-seven percent were female. Participants were equally distributed between the UK (49.5%) and the US (50.5%). A substantial proportion (42%) had completed a university degree, with 10% having a postgraduate qualification. Almost a third (29.8%) had completed either a vocational qualification or had received some undergraduate education, and 20.2% had completed high school while 5.3% had no formal qualification.

### Test materials and procedures

All participants completed a 30-question reasoning challenge, implemented as a survey on the Qualtrics platform and accessed via an anonymous link emailed to panel members. The survey

comprised 4 sub-tests described in more detail below. Each sub-test was implemented as a separate survey block and the block order was randomised. Within each block, items were presented as multiple choice or open-ended questions and their order was also randomised. The full set of questions is available alongside the study registration details (https://osf.io/7d69p/).

**Cognitive Reflection Test (CRT).** The tendency for slow analytic processing relative to fast, intuitive processing was assessed with the Revised CRT [34]. The test items represent ostensibly simple numerical reasoning problems, but the quick, intuitive answer is incorrect, and the respondent has to suppress this initial solution to produce the correct one. Results from the CRT were used to pre-screen individuals for their hypothesised susceptibility to cognitive bias. For the purposes of screening workers, we counted those as having at least 2 out of 4 questions correct as having 'passed' the CRT. We conducted a similar analysis, details of which are available in S2 Fig, restricting the sample to those workers who correctly answered at least 3 out of 4 CRT questions. Application of the more stringent criterion reduced the sample to N = 19. There is an inevitable trade-off between individual performance requirements and attrition rates. To study the effect of group size, we elected to examine a larger sample of 'sufficient' performers rather than a small sample of 'excellent' performers on the CRT.

**Raven's Standard Progressive Matrices (RSPM).** We used a nine-item, validated short-form version of the RSPM [35]. In its original form, the RSPM is one of the most commonly used intelligence tests. Each item is composed of an incomplete pattern matrix that can be completed using abstract, inductive reasoning [36]. The short form version of the test predicts the total score on the full 60-item version of the original RSPM with good accuracy and its psychometric properties are thought to be comparable [35]. Results from this subtest were used to estimate the participants' full-scale IQ.

**Heuristics-and-Biases Test (HBT).** Susceptibility to common reasoning errors was tested with nine questions from a previously published set of heuristics-and-biases tasks [28]. The items reflect important aspects of rational thinking, including methodological reasoning, sample size accounting, probability matching, covariation detection, understanding of regression to the mean, probabilistic reasoning, susceptibility to the gambler's fallacy and causal base rate errors.

**Syllogistic Reasoning Test (SRT).** This test contains eight items from a published, validated test of syllogistic reasoning [37], in which both logical validity and believability of the conclusion are manipulated. Participants are asked to evaluate whether the conclusions logically follow the statements. For half the items, the conclusion was semantically incongruent with the logical validity (i.e. the conclusion was believable, but logically invalid based on the premises, or the conclusion was unbelievable, but logically valid based on the premises). Incorrect classification of the concluding statement thus signals susceptibility to the common 'belief bias'.

## Analyses

Subtest scores were calculated by assigning a value of "1" for a correct answer and "0" for an incorrect answer and summing these values across all items for a given test. To produce comparable performance metrics across the four subtests, we present the descriptive performance data for the sample expressed in percentage correct (test score/N(items in the test) x 100). We also performed a descriptive analysis of response distributions for each of the test items in the HBT and SRT, tabling the frequency with which each response option was selected by the full sample of N = 95 participants. This enabled us to assess whether the "population" from which we derived the various nominal groups showed systematic response biases.

To explore the extent to which cognitive reflection might explain variation in susceptibility to bias, we examined bivariate (non-parametric) correlations between participants' scores on

the CRT and the HBT and SRT. We then included CRT performance in a regression model, alongside demographic factors (age, gender and whether or not the participant had completed tertiary education) and estimated intelligence (based on the RSPM score) to determine the explanatory power of cognitive reflection on bias susceptibility.

For the analysis of group performance, we selected random samples (without replacement, unweighted) from the available participant pool to create nominal 'groups' comprised of n = 1 to n = 25 participants. We describe group performance on the HBT and SRT, examining accuracy on individual items and calculating the overall test score. For each group of size n, we determined the group answer by relative majority (e.g. if n = 7 and three group members select answer "A", two group members select answer "B", and two group members select answer "C", then the group's answer is "A"). Ties were resolved by random selection (e.g. an equal number of group members selected answer "A" and answer "B", the tie was broken by randomly selecting between these two answers). The groups' overall test scores for the HBT and the SRT were determined by summing item scores (1 for a correct answer and 0 for any incorrect answer). To ensure reliability, we repeated the above sampling procedure and test score calculation 1000 times for each group size n. This results in an estimation of the likelihood of observing a correct answer for each group size n. We then plotted the curves for the relationship between group size and estimated accuracy.

## Results

### Descriptive statistics

We examined the sampling pool's performance on the reasoning tasks included in the crowd-solving challenge (Table 1) and compared the performance of the full sample to those who 'passed' the CRT screening test.

We observed moderate to strong correlations between the CRT and the cognitive bias tests, namely the HBT (r = .440, p < .001), and the SRT belief-bias items (r = .563, p < .001). Smaller (moderate) correlations were observed between the RSPM and the HBT (r = .303, p = .003), and the SRT belief-bias items (r = .370, p < .001).

The CRT score was used to split the pool of workers based on their tendency towards Cognitive Reflection. A cut-off criterion of 2/4 items answered correctly resulted in 35 workers who 'passed' the CRT and 60 workers who 'failed' the CRT. The mean CRT score for the 'passed' workers (M = 2.63, SD = .65) was significantly different from the 'failed' workers (M = .48, SD = .50), t(58.22) = 16.86, p < .001, 95% CI for the difference between the means = [1.89–2.40], suggesting that there was a meaningful difference between the two subsets. We then conducted hierarchical regressions using scores on the HBT and SRT (belief bias items

**Table 1. Descriptive statistics of performance on the screening test and the reasoning tasks.**

|  | Possible score range | Mean (SD) | Median |
|---|---|---|---|
| CRT | 0–4 | 1.27 (1.18) | 1 |
| RSPM | 0–9 | 4.76 (1.89) | 5 |
| HBT | 0–9 | 3.63 (2.02) | 3 |
| SRT–incongruent items (belief bias) | 0–4 | 1.45 (1.30) | 1 |
| SRT–congruent items (non-belief bias) | 0–4 | 3.66 (0.65) | 4 |

The table summarises the sample's (N = 95) performance on the four sub-tests of the crowdsourced reasoning challenge; the Cognitive Reflection Test (CRT), Raven's Standard Progressive Matrices (RSPM), the Heuristics and Biases Test (HBT) and the Syllogistic Reasoning Test (SRT).

**Table 2. Cognitive reflection predicts reasoning over and above demographic differences.**

| Model | Predictors | Standardised coefficient (β) | t-statistic | p-value | Adj. R² | ΔR² | ΔF | p-value |
|---|---|---|---|---|---|---|---|---|
| **Dependent: HBT Score** | | | | | | | | |
| 1 | (Constant) | | 1.800 | 0.075 | | | | |
| | Age | 0.122 | 1.086 | 0.280 | | | | |
| | Gender | 0.238 | 2.138 | 0.035 | | | | |
| | Education level | 0.024 | 0.232 | 0.817 | | | | |
| | | | | | **0.069** | **0.1** | **F(3,89) = 3.290** | **0.024** |
| 2 | (Constant) | | -2.738 | 0.007 | | | | |
| | Age | 0.171 | 1.605 | 0.112 | | | | |
| | Gender | 0.201 | 1.916 | 0.059 | | | | |
| | Education level | -0.072 | -0.718 | 0.474 | | | | |
| | Estimated_IQ | 0.354 | 3.587 | 0.001 | | | | |
| | | | | | **0.179** | **0.115** | **F(1,88) = 12.866** | **0.001** |
| 3 | (Constant) | | -1.583 | 0.117 | | | | |
| | Age | 0.100 | 0.943 | 0.348 | | | | |
| | Gender | 0.132 | 1.267 | 0.209 | | | | |
| | Education level | -0.078 | -0.809 | 0.421 | | | | |
| | Estimated IQ | 0.262 | 2.611 | 0.011 | | | | |
| | Passed CRT | 0.299 | 2.820 | 0.006 | | | | |
| | | | | | **0.239** | **0.066** | **F(1,87) = 7.954** | **0.006** |
| **Dependent: SRT score (belief bias items only)** | | | | | | | | |
| 1 | (Constant) | | 0.573 | 0.568 | | | | |
| | Age | 0.153 | 1.327 | 0.188 | | | | |
| | Gender | 0.094 | 0.830 | 0.409 | | | | |
| | Education level | 0.087 | 0.831 | 0.408 | | | | |
| | | | | | **0.028** | **0.06** | **F(3,89) = 1.881** | **0.139** |
| 2 | (Constant) | | -3.355 | 0.001 | | | | |
| | Age | 0.205 | 1.893 | 0.062 | | | | |
| | Gender | 0.055 | 0.517 | 0.607 | | | | |
| | Education level | -0.015 | -0.149 | 0.882 | | | | |
| | Estimated IQ | 0.379 | 3.779 | 0.000 | | | | |
| | | | | | **0.154** | **0.131** | **F(1,88) = 14.283** | **<0.001** |
| 3 | (Constant) | | -1.797 | 0.076 | | | | |
| | Age | 0.100 | 0.982 | 0.329 | | | | |
| | Gender | -0.047 | -0.468 | 0.641 | | | | |
| | Education level | -0.024 | -0.259 | 0.796 | | | | |
| | Estimated IQ | 0.244 | 2.520 | 0.014 | | | | |
| | Passed CRT | 0.440 | 4.306 | 0.000 | | | | |
| | | | | | **0.295** | **0.142** | **F(1,87) = 18.542** | **<0.001** |

Results from the hierarchical regression models on Heuristics and Biases Test (HBT) and Syllogistic Reasoning Test (SRT) scores.

only) as the outcome variable, and age, gender and education level (Model 1), RSPM (Model 2), and whether or not the participant passed the CRT (Model 3) as predictors. All models were statistically significant, but the CRT screening variable (i.e. whether or not the worker passed the CRT) substantially improved prediction of both HBT and SRT performance over and above the effects of demographic characteristics and intelligence (Table 2).

## Response distributions in the sampling pool

Fig 1 represents the distributions of response alternatives for the cognitive bias tests (HBT and belief bias items of the SRT), for the entire sample and the sample screened on the basis of CRT scores. To simplify the graphical representation, unique responses are not included. To test whether the screened workers were more likely to converge on the correct answer, we conducted $\chi^2$ tests on the proportion of correct/incorrect answers for those who passed and those who failed the CRT. We found a significant difference for five out of nine HBT questions and all four belief-bias SRT items (S1 Fig).

## Effect of group size on reasoning performance

We first examined overall test performance, demonstrating that nominal groups outperformed individuals on the HBT test (taking into account performance across all 9 items), regardless of whether they were screened for Cognitive Reflection. In both conditions similar curves indicate steady improvement on the HBT as nominal group size increased. The maximal performance gain observed (comparing n = 1 to n = 25), equated to a moderate to large effect size for both nominal groups taken from all workers (Cohen's d = .72), and for nominal groups taken from the workers screened for Cognitive Reflection (Cohen's d = .81). The belief-bias SRT items, on the other hand, showed a marked performance loss in nominal groups of increasing size, when we sampled from the entire population of workers (Cohen's d = .77). Among the workers screened for Cognitive Reflection, the correct answers gained a majority vote, and–in

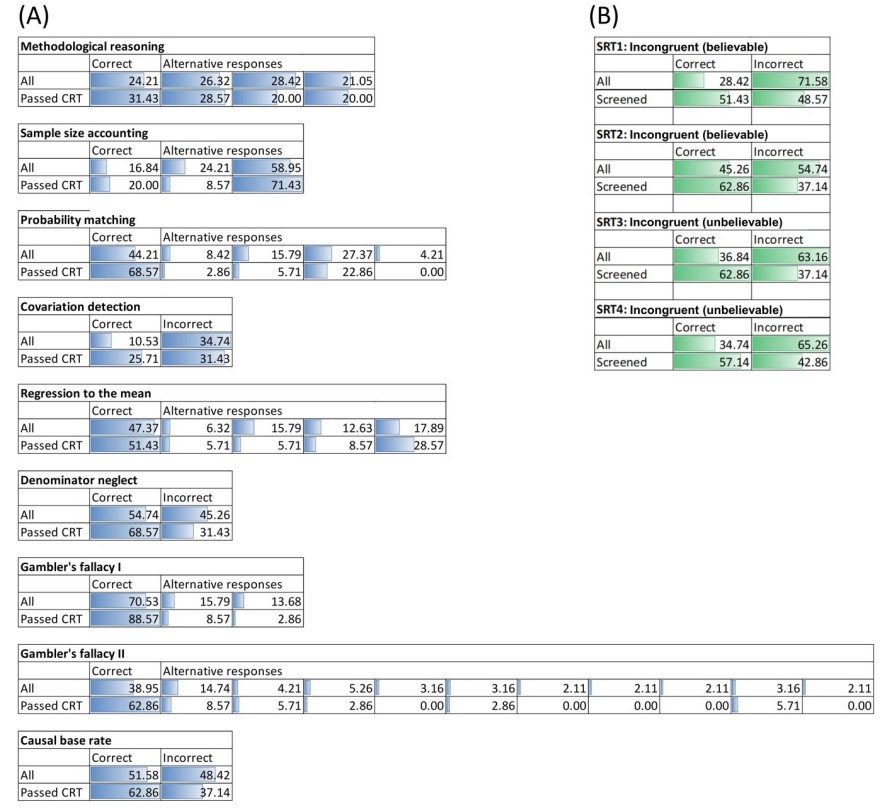

**Fig 1. Individual test item response distributions.** Panel (A) shows the response distributions for the 9 items of the Heuristics and Biases Test; panel (B) shows the response distributions for the 4 belief-bias items of the Syllogistic Reasoning Test. Results are shown for the entire sample ('all') and the sample screened on the basis of their Cognitive Reflection Test (CRT) score ('passed CRT').

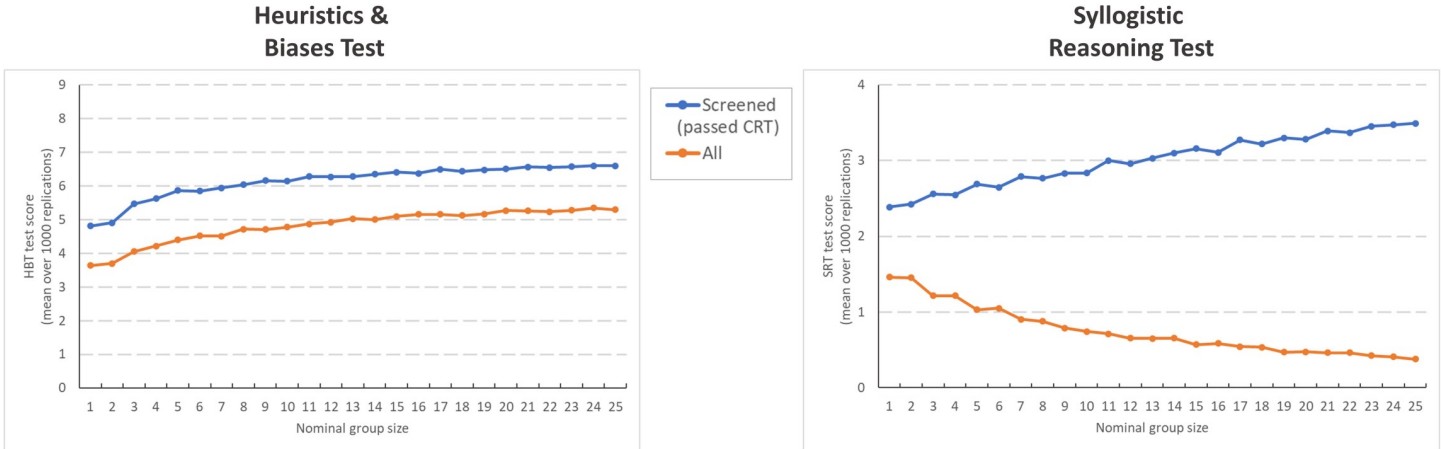

**Fig 2. Effect of nominal group size on aggregate test performance.** The relationship between test score and nominal group size for the Heuristics and Biases Test (HBT) and the Syllogistic Reasoning Test (SRT), for the entire sample ('all') and when workers were screened on the basis of their Cognitive Reflection Test (CRT) score ('passed CRT').

line with the Condorcet Jury Theorem–this shift in the response distribution (see Fig 2) also resulted in enhanced nominal group performance (Cohen's d = .55).

The link between response distribution and the effect of aggregation in nominal groups can only be fully appreciated when examining nominal group performance on individual test items (Fig 3). Our results show bias amplification for those items where the modal answer in the population (i.e. the entire sample from which participants were drawn) was incorrect (i.e. the result of a systematic cognitive bias). Considering the entire sample of workers, this occurred on the HBT items 'sample size accounting', 'covariation detection' and to a lesser extent the item 'methodological reasoning'. As can be seen in Fig 1, for each of these items, there was an incorrect response alternative that attracted a relative majority of respondents. The more pronounced the bias (the greater the majority for a biased response), the more the bias was amplified through aggregation (e.g. compare 'covariation detection' to 'methodological reasoning'). When group members were screened for Cognitive Reflection, aggregation resulted in performance gains for seven out of nine items of the HBT, and in perfect performance (100% of the re-sampled nominal groups were correct) in six out of nine items. Without screening on the other hand, nominal groups only reached near-perfect performance in three out of nine tasks. It also took larger groups to achieve this level of accuracy. The difference between screened and non-screened nominal groups was more pronounced on the SRT. Across all four belief-bias items, the nominal groups screened for Cognitive Reflection outperformed those who had not been screened. In the latter case, aggregation amplified the bias within the population from which groups were sampled, but in the former, absence of such a bias resulted in consistent performance gains as groups increased in size.

## Discussion

Quality control is a significant challenge in many crowdsourcing ventures, but the impact of cognitive bias is relatively understudied in this field. The aim of our study was to assess the susceptibility of online workers to common cognitive biases and to test the extent to which individual differences in cognitive reflection could be leveraged to enhance the output quality in 'crowdsolving' challenges. We used data collected from an online survey panel who completed a range of reasoning challenges.

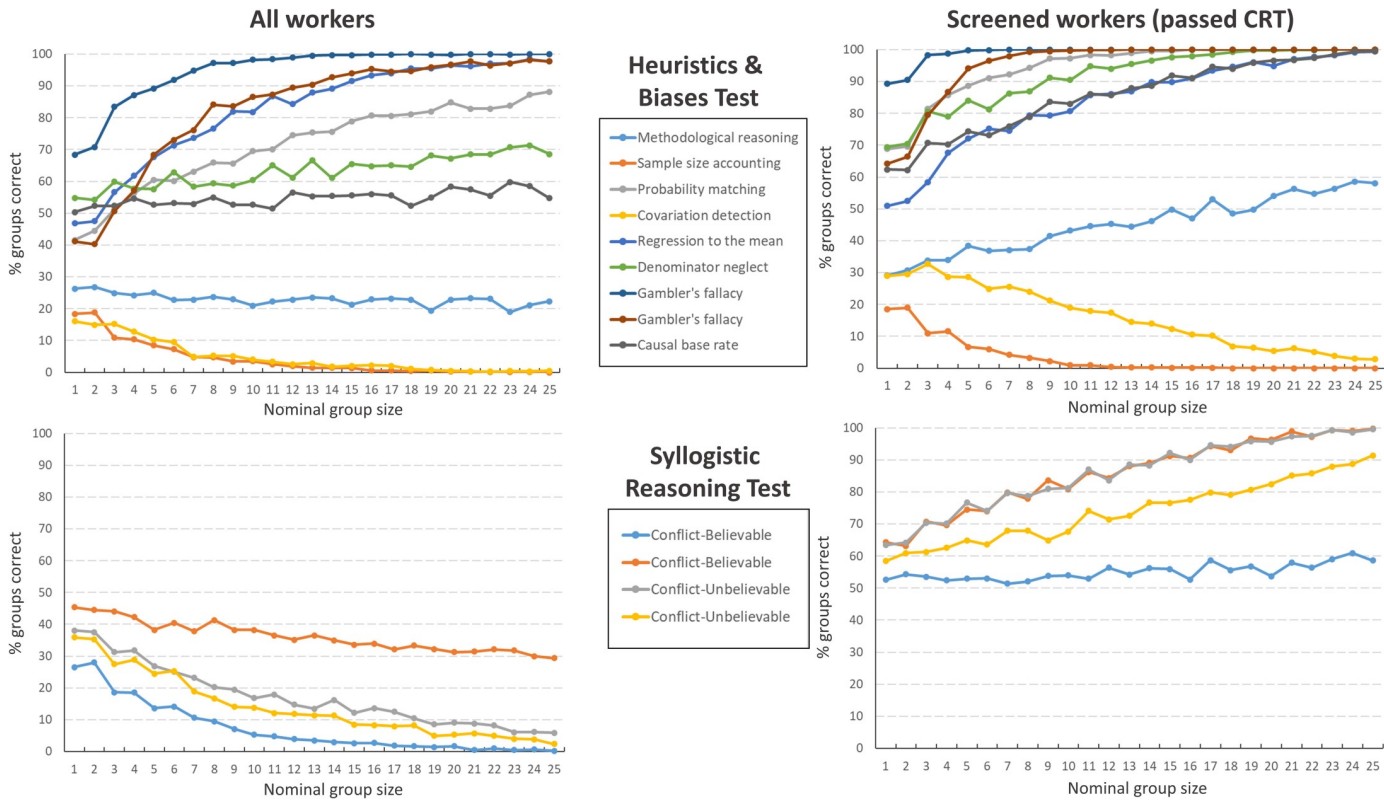

**Fig 3. Effect of nominal group size on all items.** The relationship between group accuracy and nominal group size for individual test items on the Heuristics and Biases Test (HBT) and the Syllogistic Reasoning Test (SRT), when workers were or were not screened for cognitive reflection prior to group formation.

The Condorcet Jury Theorem predicts that when the members of a group are more likely to be right than wrong, then their collective answer (by plurality vote), is also more likely to be right than wrong. Furthermore, the probability that the group arrives at the correct response increases with the size of the group [25]. This principle thus supports the use of simple aggregation as a method for harnessing collective wisdom. However, some of the most interesting applications of crowdsourcing are likely to trigger cognitive biases that may produce a systematic deviation from 'the truth'. Whether these intuitively appealing, yet incorrect responses are salient enough to achieve a relative majority in a crowdsourced sample is the first empirical question we investigated here.

## Do reasoning errors lead to bias amplification in online panels?

The online workers we sampled showed similar overall performance levels compared to previous studies asking students the same heuristic and biases questions (average response accuracy was 40.4% for the HBT, compared to 48.1%, as reported in [28]). Although the error rate varied considerably among questions (29.5% - 85.3%), as a set, these reasoning challenges can universally be considered taxing and online workers are, as demonstrated here, prone to mistakes. However, we found that poor performance on the HBT was generally not due to systematic error. On most items, workers supported a range of different response alternatives, amongst which the normatively correct answer was sufficiently salient to attract a relative majority. We observed only a small subset of HBT items on which the crowdsourced sample preferred a

unique incorrect response alternative, which would be indicative of a commonly committed systematic reasoning error: the workers typically discounted sample size when assigning a probability to an event and they also lacked consideration of statistical information when making judgements about the relatedness of two phenomena.

Second, the SRT requires a binary judgement as to whether a conclusion, given two premises, is logically valid. The average error rate of 63.7% demonstrates that the majority of workers failed to ignore meaning when assessing the conclusion, showing the well-established belief-bias effect [38–40]. The over-reliance on prior beliefs when assessing novel information can be particularly detrimental to evidence-based decision making, as might be required in crowdsolving challenges. While the source of the belief bias effect has been debated (e.g. see [41] and more recently [42, 43]), what matters here is that whatever mechanism is responsible, a consistent pattern of erroneous responding can be observed in our crowdsourced sample; none of the incongruent SRT items commanded a majority for the correct answer. The higher incidence of systematic error in the SRT compared to the HBT may stem from the facility with which a cognitive shortcut can be applied and the number of available response options. The realistic decision context vignettes of the HBT are open to various interpretations and multiple erroneous solutions, differentially arising from cognitive or response bias, inattention or lack of effort. In the forced-choice SRT however, inattention, misinterpretation and systematic bias would all lead to the same incorrect outcome, thus resulting in a more systematic deviation from the truth.

Having established the overall reasoning performance in our base sample of online workers, we examined collective solutions by aggregating individual workers' responses. For most of the HBT reasoning challenges, we found that, in accordance with the Generalized Condorcet Jury Theorem, that groups of increasing size were more likely to home in on the correct solution. Where each respondent's probability of choosing the correct option from among k options is marginally above 1/k and the probability of choosing incorrect ones just below that, the probability of the correct option being selected by the nominal group increases more slowly with group size [25]. In line with this, we observed different group size–group performance curves due to variation in the extent to which the HBT reasoning challenges elicited polarised or more distributed response preferences. For instance, the 'denominator neglect' question, which showed an almost even distribution of response preferences (slightly favouring the correct answer), had a marginal improvement in accuracy with increasing group size. The 'regression to the mean' question, which had a lower accuracy rate, but also elicited a range of incorrect solutions, showed a steep increase in accuracy with group size. The theorem of course works in the other direction, too, as illustrated by the SRT data; when the population is consistently more likely to be wrong than right, the probability of the collective choice being wrong dramatically increases with group size.

Overall, these results demonstrate empirically that, while pervasive cognitive biases may not be as prevalent as expected in a crowdsourced sample, systematic errors limit the utility of the nominal group technique for harnessing the wisdom of the crowd. Statistical aggregation can be detrimental, but only when a majority within the population from which responses are sampled can be "duped" into selecting a single intuitive or alluring alternative. It also highlights that the capacity for enhanced reasoning performance (and the risk of bias amplification) using a simple crowdsourcing approach depends on the specific demands of the reasoning challenge. A task requester may therefore wish to consider what kinds of biases and heuristics are likely to be generated by a specific task, and how prevalent these may be in the population of interest. However, this is likely to be an unknown factor, which suggests that additional attempts at mitigating bias may be warranted.

## Harnessing individual differences to enhance collective reasoning

The second question we wished to address was the extent to which a simple quality-control check could be implemented to screen workers to achieve a more favourable response distribution. That is, if we can identify individuals less susceptible to cognitive bias and reasoning error, the probability of individuals supporting the correct answer (or at least not collectively supporting a single incorrect answer) would, in theory, result in improved nominal group performance as per the Condorcet Jury Theorem. We used the Cognitive Reflection Test (CRT), a measure known to be associated with improved performance on challenging reasoning tasks and real-life risk taking behaviour [28, 29], as a screening tool for identifying potentially high-performing workers.

About 38% of our original pool of workers met our screening criterion (answering at least 2/4 CRT questions correctly). This subset of workers was significantly more accurate (Fig 4) and showed a stronger response preference for the correct answer (S1 Fig). We then verified whether this performance improvement at the individual level was sufficient to produce the desired effects of group aggregation. We found that nominal groups composed of screened workers outperformed nominal groups unscreened for Cognitive Reflection on all but one of the Heuristics and Biases Tests, and on all of the incongruent (belief-bias) syllogistic reasoning items. Furthermore, compared to our wider sample, the screened sample required smaller group sizes to reach the same accuracy. The performance gain was most prominent for the SRT, where the bias amplification observed in the unscreened nominal groups was entirely mitigated in the screened sample.

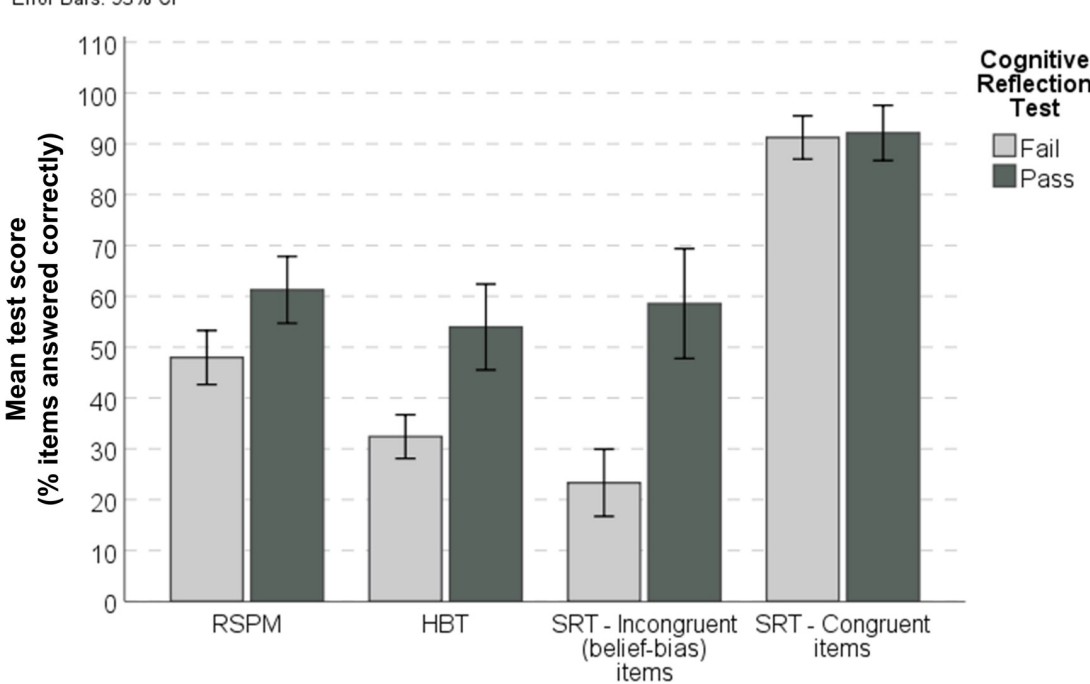

**Fig 4. Reasoning task performance is associated with cognitive reflection.** The shows the mean percentage correct on the Heuristics and Biases Test (HBT), the 2 parts of the Syllogistic Reasoning Test (SRT), and the abbreviated Ravens Standard Progressive Matrices (RSPM) for the workers (n = 35) who 'passed' the Cognitive Reflection Test (CRT) and those who did not (n = 60).

## Limitations and future research

While our results point to a general benefit of pre-screening workers for their tendency to override an incorrect 'gut' response and engage in further reflection, establishing the external validity and practical utility of this approach requires further work. Firstly, we used an online labour market accessed through the popular Qualtrics survey platform. Our relatively small base sample from which the nominal groups were drawn may not be representative of all labour markets. Replicating this finding across different platforms would help in establishing the utility of this approach. In addition, because distribution of the task is outsourced to a third party (i.e. Qualtrics), the task requester does not have access to information about non-respondents. In this instance, without further knowledge about the causes of the non-response rates, our ability to draw conclusions about repeatability of the results and the external validity of the findings is limited.

Secondly, the cross-sectional nature of the approach described here means that we cannot draw causal links between performance on the screening test and reasoning performance of individuals or aggregates. There are a range of other factors and moderating variables, not included in our analysis, that could influence this relationship, including individual (e.g. personality traits) and contextual (e.g. distractions during task performance). However, from a practical point of view, our results demonstrate that a simple screening step can be effective. Indeed, for a task requester who is concerned with accuracy (solving a problem correctly), speed (getting an expeditious conclusion) and cost (not paying for additional responses that are low quality), rapid screening may provide an appropriate shortcut. It also avoids the need to ask more intrusive or sensitive questions that tap into the personal domain of online workers (e.g. demographic data, occupation, educational history). These data are often available from services like Qualtrics and Prolific that provide access to labour markets, but using them to pre-select workers attracts additional, potentially significant fees.

Thirdly, the quick completion of the CRT (our workers took on average 2'14") suggests it is an efficient assessment. However, the added cost and time of screening out highly bias-susceptible workers needs to be traded off against overall gains in efficacy, i.e. requiring fewer crowd-sourced submissions to achieve the same or a superior output. It is not a trivial task to derive specific guidance about the number of workers that would be required to state with confidence that an unbiased and well-reasoned answer will be achieved. Furthermore, pre-screened groups did not unequivocally produce an entirely unbiased, well-reasoned correct answer (although they certainly showed improved performance), which adds a degree of uncertainty that would need to be taken into account in a cost-benefit analysis.

Finally, these results provide initial empirical substantiation of a general principle, on a small set of experimental questions that are not representative of the spectrum of potential crowdsolving challenges. Nevertheless, the findings are in line with previous research suggesting that cognitive reflection capabilities are broadly predictive of susceptibility to cognitive bias, affecting real-life reasoning skills [29]. Our study is the first to demonstrate a similar trend in online labour markets and highlights its practical implications in this context. The true potential of pre-screening in this manner will require further validation with more complex and realistic outcome measures that are of interest to requesters in online labour markets.

## Conclusion

Previous research has demonstrated that: (1) people regularly, and in predictable ways, deviate from normatively correct solutions when solving reasoning tasks; (2) individuals' propensity towards cognitive reflection correlates with their susceptibility to systematic reasoning biases; and (3) when individuals' probability of choosing the correct option from among k options is

below 1/k and the probability of choosing incorrect ones just above that, the probability of the correct option being selected by the group decreases with group size. The study we report offers the first look into the connections between these three separate research agendas. We show that online labour markets may not exhibit strong systematic reasoning biases, and as a consequence their collective output generally improves with group size. When bias is suspected however, assessing workers on their cognitive reflection capacity proves to be an effective pre-screening tool for obtaining higher-quality crowdsolved answers.

## Supporting information

**S1 Fig. $\chi^2$ tests comparing the proportion of workers supporting the correct answer between two subsets of workers: Those who failed the Cognitive Reflection Test (CRT) vs those who passed the CRT.**
(DOCX)

**S2 Fig. Graphs depicting the relationship between group size and nominal group accuracy when workers were screened on the Cognitive Reflection Test (CRT) using a more stringent performance criterion.** We screened N = 19 workers who answered at least 3 of the 4 CRT questions correctly and these workers were thus singled out for comparison against the full sample. As a result, the maximum nominal group size we could investigate was 19. Panel (A) shows the overall performance of nominal groups on the Heuristics and Biases Test and the Syllogistic Reasoning Test; panel (B) shows the item-by-item performance of nominal groups on the Heuristics and Biases Test and the Syllogistic Reasoning Test.
(DOCX)

## Acknowledgments

We thank Neil R. Thomason for helpful discussions and comments on an earlier version of this manuscript.

## Author Contributions

**Conceptualization:** Ans Vercammen, Mark Burgman.

**Data curation:** Ans Vercammen.

**Formal analysis:** Ans Vercammen, Alexandru Marcoci.

**Funding acquisition:** Mark Burgman.

**Investigation:** Ans Vercammen, Alexandru Marcoci.

**Methodology:** Ans Vercammen.

**Project administration:** Ans Vercammen, Mark Burgman.

**Resources:** Ans Vercammen.

**Supervision:** Mark Burgman.

**Validation:** Ans Vercammen, Alexandru Marcoci.

**Visualization:** Ans Vercammen.

**Writing – original draft:** Ans Vercammen.

**Writing – review & editing:** Ans Vercammen, Alexandru Marcoci, Mark Burgman.

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
