## [Decision Letter · Decision Letter 0]

25 Feb 2021

PONE-D-20-37862

Pre-screening workers to overcome bias amplification in online labour markets

PLOS ONE

Dear Dr. Vercammen,

Thank you for submitting your manuscript to PLOS ONE. After careful consideration, we feel that it has merit but does not fully meet PLOS ONE’s publication criteria as it currently stands. Therefore, we invite you to submit a revised version of the manuscript that addresses the points raised during the review process.

The revised version should take into account all the comments stated in the reports.

We look forward to receiving your revised manuscript.

Kind regards,

Petri Böckerman

Academic Editor

PLOS ONE

Journal Requirements:

Reviewers' comments:

Reviewer's Responses to Questions

**Comments to the Author**

1. Is the manuscript technically sound, and do the data support the conclusions?

Reviewer #1: Yes

Reviewer #2: Yes

2. Has the statistical analysis been performed appropriately and rigorously? 

Reviewer #1: Yes

Reviewer #2: Yes

3. Have the authors made all data underlying the findings in their manuscript fully available?

Reviewer #1: Yes

Reviewer #2: No

4. Is the manuscript presented in an intelligible fashion and written in standard English?

Reviewer #1: Yes

Reviewer #2: Yes

5. Review Comments to the Author

Reviewer #1: Hello, Its honor for me to review your research manuscript . Please improve the result & discussion part . This part need improvement further and also improve the conclusion part . I hope you will improve it

Reviewer #2: Comments

1. The revised introduction should state the contribution of the paper to the earlier literature.

2. The sample size is relatively small (page 6). What is the external validity of the findings that are presented in the paper?

3. The description of the data should be improved. Was non-response to the survey random or not? This information would be useful in order to better understand the estimation results that are presented in the paper.

4. Do the data contain (survey) weights? Are they used in the estimations or not?

5. The statistical analyses do not address causal questions. Therefore, the paper reports (conditional) correlations between the variables of interest using cross-sectional variation. But the measures that are used in the empirical specifications are subjective. This implies that the unobserved individual-level characteristics such as personality traits may have a substantial influence on all variables that are used in the analysis. This limits the conclusions that can be drawn from the estimates.

6. The concluding section could discuss more about the practical lessons that stem from the results.

6. PLOS authors have the option to publish the peer review history of their article (what does this mean?). If published, this will include your full peer review and any attached files.

Reviewer #1: No

Reviewer #2: No

---

## [Author Response · Author response to Decision Letter 0]

3 Mar 2021

Below we provide our point-by-point response to the reviewers’ comments.

1. Is the manuscript technically sound, and do the data support the conclusions?

Reviewer #1: Yes

Reviewer #2: Yes

RESPONSE:

Thank you. To answer Reviewer#2’s queries with regard to the external validity and causality of the findings (mentioned below), the revised manuscript now clearly outlines the limitations of the chosen design and the implications of this for future research [Lines 476-486 ]

2. Has the statistical analysis been performed appropriately and rigorously? 

Reviewer #1: Yes

Reviewer #2: Yes

RESPONSE:

Thank you. No further changes have been made to the analyses.

3. Have the authors made all data underlying the findings in their manuscript fully available?

Reviewer #1: Yes

Reviewer #2: No

RESPONSE:

The full dataset and other relevant information such as the analysis code and the experimental materials are provided in full and referenced in the methods section [Line 132].

4. Is the manuscript presented in an intelligible fashion and written in standard English?

Reviewer #1: Yes

Reviewer #2: Yes

RESPONSE:

Thank you. We have ensured that our subsequent edits also meet the expected English language standard.

5. Review Comments to the Author

Reviewer #1: Hello, Its honor for me to review your research manuscript . Please improve the result & discussion part . This part need improvement further and also improve the conclusion part . I hope you will improve it

RESPONSE:

The reviewer did not specify which aspects of the results, discussion or conclusion required polishing. However, based on this advice and on the questions raised by the second reviewer, we have made revisions to highlight the major limitations of the work, explore future research opportunities, and identify potential real-life implications of our findings, without extrapolating beyond the bounds of what the results indicate. We hope this addresses Reviewer#1’s queries and satisfies Reviewer2. 

Reviewer #2: Comments

1. The revised introduction should state the contribution of the paper to the earlier literature.

RESPONSE:

We have clarified that our paper provides empirical verification based on theoretical assumptions about the occurrence of reasoning errors in individuals and the effects of aggregation in crowdsourcing/crowdsolving. We point out that these assumptions have not yet been tested specifically in a crowdsolving setting where randomly selected individual members of an online panel provide independent results that are aggregated post-hoc. While it can be assumed that cognitive biases are just as prevalent in these types of samples and situations, this is an untested assumption, and, more importantly, we also do not know how group size affects collective performance when erroneous thinking/responding is common in individuals. We have made this point clearer in the introduction. The second unresolved issue that we address here is whether a simple strategy that has been demonstrated to effectively identify “effortful thinkers” could be applied to mitigate the effects of cognitive bias in crowdsolving. We now emphasise these specific novel aspects of our paper more clearly in the revised manuscript [Lines 98-110].

2. The sample size is relatively small (page 6). What is the external validity of the findings that are presented in the paper?

RESPONSE:

We agree with the reviewer that this is an important issue. However, compared to other papers (e.g. the seminal paper on crowdsourced collective intelligence by Kosinski et al 2012), our study has a larger base sample. To ensure reliability of our sample size analysis, we repeated the sub-sampling procedure and test score calculation 1000 times for each group size n. Of course, we cannot claim that the 105 individuals that formed our base sample are entirely representative of “online workers”. However, we did not specify any restrictions on the selection of individuals by the Qualtrics panel service, and thus accessed a pool of participants that are broadly representative of casual online labourers. We reflect on the implications of this approach in the revision [Lines 435-439].

3. The description of the data should be improved. Was non-response to the survey random or not? This information would be useful in order to better understand the estimation results that are presented in the paper.

RESPONSE:

Because of the way these online labour markets work, the requester does not have access to information about non-responses. In short, we cannot provide data on whether non-response was random or not. It would be interesting, from a theoretical perspective, to understand how this form of sampling bias might have affected nominal group performance. However, from a practical point of view, our approach reflects the reality of online labour markets. This consideration also speaks to the issue of our base sample’s representativeness, which we have discussed above. We expand on these issues in our limitations section [Lines 435-443].

4. Do the data contain (survey) weights? Are they used in the estimations or not?

RESPONSE:

No, the survey data was not weighted in our analyses. We now specify this in the methods section [Line 138]. Of course, one could view excluding respondents on the basis of the screening test as an extreme form of weighting (i.e. survey weight = 0 if they did not pass the screening test). Logically, one other approach could be to assign a weight to individual responses on the basis of some indicator variable (e.g. score on the screening test, education level or another way of classifying reasoning ability). However, our aim was to examine baseline nominal group performance (i.e. without preselection or intervention) and then to test a simple and cost-effective strategy for quickly “weeding out” responses that would unduly and negatively affect nominal group performance. Weighting is a more resource intensive and – from a task requester perspective – costly option to achieve this goal. It was beyond the scope of the current investigation to address this.

We note that we provide a full description and a copy of the raw data, and it is therefore open to any further analyses and exploration by interested parties.

5. The statistical analyses do not address causal questions. Therefore, the paper reports (conditional) correlations between the variables of interest using cross-sectional variation. But the measures that are used in the empirical specifications are subjective. This implies that the unobserved individual-level characteristics such as personality traits may have a substantial influence on all variables that are used in the analysis. This limits the conclusions that can be drawn from the estimates.

RESPONSE:

We agree with the reviewer that the nature of the study does not allow us to draw causal links, and that there are traits or characteristics that we did not measure that might affect the performance of online workers. However, from a practical perspective causality is not a prerequisite. That is, we demonstrate that using the CRT to screen workers could be a simple and easy tool to mitigate against the negative influence of biased reasoning in a general sample of online workers. Task requesters will rarely have a detailed understanding of the characteristics of their online workforce. More elaborate subsampling on the basis of education, personality traits etc. would be more arduous and costly than using the CRT, which provides a shortcut to optimising performance that is reasonable, simple to implement and does not require the worker to divulge more sensitive or personal information (e.g. about their personality, their educational history, their occupation). 

Nevertheless, we agree that as we have tested the applicability of this approach on a small set of reasoning tasks, the broader use of such a shortcut remains to be demonstrated [Lines 468-473]. Indeed, in other types of challenges, some of the variables mentioned by the reviewer may be more important or may play a moderating role. We have included this consideration in our revised limitations section [Lines 444-448 ]

6. The concluding section could discuss more about the practical lessons that stem from the results.

RESPONSE:

We thank the reviewer for this suggestion and have made the requested edits, considering the findings from the perspective of a task requester using online labour markets [Lines 448-455].

---

## [Decision Letter · Decision Letter 1]

10 Mar 2021

Pre-screening workers to overcome bias amplification in online labour markets

PONE-D-20-37862R1

Dear Dr. Vercammen,

We’re pleased to inform you that your manuscript has been judged scientifically suitable for publication and will be formally accepted for publication once it meets all outstanding technical requirements.

Kind regards,

Petri Böckerman

Academic Editor

PLOS ONE

Additional Editor Comments (optional):

I am happy with the revised version of the paper. I like the research question, the structure of the paper, the quality of writing, and the way the authors describe their empirical proceeding and results. Most importantly, the authors have addressed all the issues stated in my referee report for the first version appropriately.

Reviewers' comments:

Reviewer's Responses to Questions

**Comments to the Author**

1. If the authors have adequately addressed your comments raised in a previous round of review and you feel that this manuscript is now acceptable for publication, you may indicate that here to bypass the “Comments to the Author” section, enter your conflict of interest statement in the “Confidential to Editor” section, and submit your "Accept" recommendation.

Reviewer #1: All comments have been addressed

Reviewer #2: All comments have been addressed

2. Is the manuscript technically sound, and do the data support the conclusions?

Reviewer #1: Yes

Reviewer #2: Yes

3. Has the statistical analysis been performed appropriately and rigorously? 

Reviewer #1: Yes

Reviewer #2: Yes

4. Have the authors made all data underlying the findings in their manuscript fully available?

Reviewer #1: Yes

Reviewer #2: No

5. Is the manuscript presented in an intelligible fashion and written in standard English?

Reviewer #1: Yes

Reviewer #2: Yes

6. Review Comments to the Author

Reviewer #1: I am glad to review and assess this interesting article, The Instruction section, literature part, and methodology portions are adequate. I suggest the authors improve the Literature section by adding some latest articles' citations to enhance the work quality.

Overall, the manuscript is a good piece of work. I recommend that authors do a little more work and add the latest literature to support the study, as suggested. The English level is good and smooth, e.g., the language standard, specifically the grammar, of sufficient quality to meet scientific merit for publication. I accept this manuscript , as I have recommended.

Reviewer #2: See Additional Editor Comments (above)

7. PLOS authors have the option to publish the peer review history of their article (what does this mean?). If published, this will include your full peer review and any attached files.

Reviewer #1: No

Reviewer #2: Yes- Petri Böckerman

---

## [Editor Report · Acceptance letter]

15 Mar 2021

PONE-D-20-37862R1 

Pre-screening workers to overcome bias amplification in online labour markets 

Dear Dr. Vercammen:

I'm pleased to inform you that your manuscript has been deemed suitable for publication in PLOS ONE. Congratulations! Your manuscript is now with our production department. 

Kind regards, 

on behalf of

Professor Petri Böckerman 

Academic Editor

PLOS ONE